# Intertwining Neuropathogenic Impacts of Aberrant Circadian Rhythm and Impaired Neuroregenerative Plasticity in Huntington's Disease: Neurotherapeutic Significance of Chemogenetics

Sowbarnika Ravichandran [1,2,†], Ramalingam Suhasini [3], Sudhiksha Madheswaran Deepa [1,2], Divya Bharathi Selvaraj [1], Jemi Feiona Vergil Andrews [1], Viruthachalam Thiagarajan [3,4,*] and Mahesh Kandasamy [1,2,4,*]

1 Laboratory of Stem Cells and Neuroregeneration, Department of Animal Science, School of Life Sciences, Bharathidasan University, Tiruchirappalli 620024, India
2 School of Life Sciences, Bharathidasan University, Tiruchirappalli 620024, India
3 Photonics and Biophotonics Lab, School of Chemistry, Bharathidasan University, Tiruchirappalli 620024, India
4 Faculty Recharge Programme, University Grants Commission (UGC-FRP), New Delhi 110002, India
* Correspondence: vthiags@gmail.com or v.thiagarajan@bdu.ac.in (V.T.); pkmahesh5@gmail.com or mahesh.kandasamy@bdu.ac.in (M.K.)
† Current address: Institute of Anatomy, Johannes Gutenberg University Mainz, 55128 Mainz, Germany.

**Abstract:** Huntington's disease (HD) is a progressive neurodegenerative disorder characterized by abnormal progressive involuntary movements, cognitive deficits, sleep disturbances, and psychiatric symptoms. The onset and progression of the clinical symptoms have been linked to impaired adult neurogenesis in the brains of subjects with HD, due to the reduced neurogenic potential of neural stem cells (NSCs). Among various pathogenic determinants, an altered clock pathway appears to induce the dysregulation of neurogenesis in neurodegenerative disorders. Notably, gamma-aminobutyric acid (GABA)-ergic neurons that express the vasoactive intestinal peptide (VIP) in the brain play a key role in the regulation of circadian rhythm and neuroplasticity. While an abnormal clock gene pathway has been associated with the inactivation of GABAergic VIP neurons, recent studies suggest the activation of this neuronal population in the brain positively contributes to neuroplasticity. Thus, the activation of GABAergic VIP neurons in the brain might help rectify the irregular circadian rhythm in HD. Chemogenetics refers to the incorporation of genetically engineered receptors or ion channels into a specific cell population followed by its activation using desired chemical ligands. The recent advancement of chemogenetic-based approaches represents a potential scientific tool to rectify the aberrant circadian clock pathways. Considering the facts, the defects in the circadian rhythm can be rectified by the activation of VIP-expressing GABAergic neurons using chemogenetics approaches. Thus, the chemogenetic-based rectification of an abnormal circadian rhythm may facilitate the neurogenic potentials of NSCs to restore the neuroregenerative plasticity in HD. Eventually, the increased neurogenesis in the brain can be expected to mitigate neuronal loss and functional deficits.

**Keywords:** Huntington's disease; circadian rhythm; clock genes; adult neurogenesis; chemogenetics

## 1. Introduction

Huntington's disease (HD) is an autosomal dominant hereditary neurodegenerative disorder that affects the structure and functions of the basal ganglia of the brain [1]. The progressive degeneration of gamma-aminobutyric acid (GABA)-ergic medium spiny (MSN) neurons in the brains of subjects with HD has been attributed to the expansion of polyglutamine (poly Q) segments in the huntingtin (HTT) protein resulting from more than 40 CAG repeats in the exon1 of the *HTT* gene [2–4]. Clinically, HD has been characterized by abnormal involuntary movements, neurocognitive impairments, and psychiatric

disturbances [5]. In addition, abnormal sleep–wake cycles accounting for the abnormal circadian rhythm have been identified as non-motor clinical symptoms of HD [6]. Around 90% of HD subjects have been reported to suffer from sleep disturbances [7]. Chronic sleep disturbances appear to be detrimental to the neuroplasticity responsible for neurocognitive functions [8]. Ample research reports indicate that the occurrence of neurogenesis in the hippocampus in the brains of healthy subjects contributes to learning, memory, and mood [9]. Whereas neurogenic failure in the hippocampus has been considered an underlying cellular basis of neurocognitive decline in many neurodegenerative disorders, including HD [10,11]. While the expression of mutant *HTT* gene causes aberrant gliogenic events, the neurogenic potential of neural stem cells (NSCs) and survival of new-born neurons in different brain regions including the hippocampus have been reported to be drastically impaired in experimental models of HD and post-mortem human HD brains [12–17]. The underlying molecular mechanism for the impaired proliferative and differentiation potentials of NSCs in HD brains remains obscure. In a physiological state, circadian clock genes play important roles in the regulation of NSC-derived neurogenesis, whereas impairment in the neurogenic process has been linked to the irregular circadian clock pathway [18,19]. The expression of the mutant *HTT* gene interrupts the regulation and functions of the clock genes, thereby leading to the aberrant circadian rhythm in HD [7,20,21]. Therefore, the abnormal regulation of hippocampal neurogenesis and an irregular circadian rhythm may overlap and can collectively contribute to intertwining pathogenicity leading to psychiatric disturbances and cognitive deficits in HD. Considering the facts, it can be proposed that the reversal of an irregular circadian rhythm might contribute to repair mechanisms of the brain and regenerative plasticity in HD. Therefore, the identification of the prominent molecular pathway and cellular system involved in the regulation of circadian rhythm could serve as a potential therapeutic target in HD.

GABAergic vasoactive intestinal peptide (VIP)-expressing neurons in the suprachiasmatic nucleus (SCN) of the hypothalamus play a key role in the regulation of circadian rhythm. Neuropathogenic events mediated degeneration or functional defects in the VIP neurons of SCN and improper sensory inputs can trigger abnormal circadian rhythmicity in various brain diseases [22–24]. These VIP neurons play an important role in the control of GABAergic transmission responsible for the synaptic plasticity of the pyramidal neurons in the hippocampus [25,26]. Thus, the dysregulation of GABAergic transmission resulting from the mutant HTT protein might overlap with the altered expression and functions of VIP leading to neuroregenerative failure in HD. Therefore, the implementation of therapeutic strategies that aid in the restoration or activation of VIP neurons in the brain could contribute to rectifying sleep disorder in HD. Chemogenetics has been established as a potent molecular tool to specifically regulate the intracellular-signaling pathways in tissue and organs [27,28]. The chemogenetic-based approaches provide hope to mitigate the abnormal circadian clock pathways which may be coupled with improving neuroregeneration in the brain [29]. Therefore, this article describes the potential overlap between the pathogenicity responsible for altered sleep–wake patterns and aberrant neurogenesis noticed in the brains of subjects with HD, and emphasizes the chemogenetic activation of VIP-positive GABAergic neurons in the brain as a therapeutic strategy to rectify the aberrant clock gene pathways by which neurodegenerative failure is expected to be reversed in HD.

## 2. Regulation of Circadian Rhythm in Physiological State

Circadian rhythm represents a biological chronometer of the living system that regulates, intertwines, overlaps, and synchronizes various physiological, biochemical, cellular, and genetic events, in response to the gut–brain axis, atmospheric temperature, and different sensory inputs from light and dark conditions [30–32]. In mammals, the periodic modulation of circadian rhythm has been tightly linked to both photic and non-photic stimuli [33]. In the eyes, retinal ganglion cells (RGCs) express the photopigment known as melanopsin, a key photoreceptor that mediates the non-image-forming functions of the

light and pupillary light reflexes [34]. Similarly, to rod and cone cells, RGCs are also intrinsically photosensitive units that play a key role in transmitting photic signals from the eyes to the SCN through optic chiasma [35] (Figure 1). In mammals. the master clock for circadian rhythm is positioned in the SCN that synchronizes the regulation of neuroplasticity with the daily variation of the photic signals and nonphotic inputs. The SCN is compartmentalized into the dorsal shell and ventral core subdivisions and receives inputs from three afferent pathways, namely the retinohypothalamic tract (RHT), the genicular-hypothalamic tract (GHT), and a compact serotonergic plexus of the raphe nucleus (RN) [36] (Figure 1).

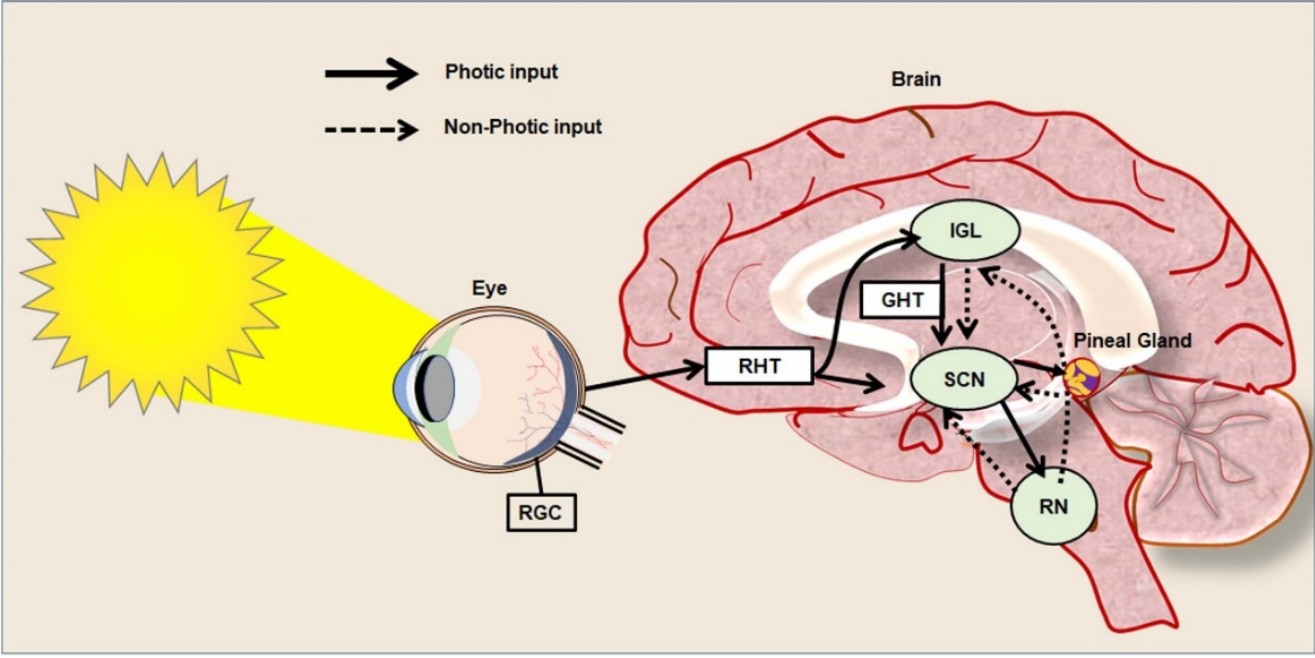

**Figure 1.** Photic and non-photic input of circadian rhythm in the healthy brain.

The digital diagram represents the photic and non-photic inputs from the retina to the brain and neural pathways among the hypothalamus, pineal gland, and raphe nucleus (RN) that regulate circadian rhythm in a healthy brain. The straight line represents the photic inputs and the dotted line represents the non-photic inputs. The photic information is generally transmitted to the suprachiasmatic nucleus (SCN) both directly, as well as indirectly from the retina. While the retinohypothalamic tract (RHT) is directly involved in the transfer of light-based input from the retina to the SCN, the retinorecipient intergeniculate leaflet (IGL) and geniculohypothalamic tract (GHT) play an indirect role in the regulation of circadian rhythm.

Eventually, the efferent projections of the SCN target the pineal gland through the VIP [37]. VIP is a major neuropeptide that is widely expressed in the gut, pancreas, and brain [38]. In particular, VIP neurons are highly present in various regions of the brain, including the cerebral cortex, amygdala, septum, hippocampus, thalamus, and hypothalamus [39,40]. VIP acts through VPAC1 and 2 receptors to stimulate the secondary messengers Cyclic adenosine 3′,5′-monophosphate (cAMP), and protein kinase A (PKA) signaling cascade, and presynaptically enhance gamma-aminobutyric acid (GABA) release in the neuronal population brain [40]. VIP receptors are widely present in the GABAergic interneurons of the hippocampus and VIP-mediated enhancement of synaptic transmission to cornu Ammonis (CA) 1 pyramidal cells involves the inhibition of GABAergic interneurons that controls the synaptic plasticity of the pyramidal neurons [40]. In the SCN, VIP neurons present contribute an important role in synchronizing the circadian cycle [41]. VIP-secreting neurons are mainly located in the ventrolateral area of the SCN, which receives the environmental input from the optic chiasm through the retinohypothalamic tract and

plays a vital role in regulating the circadian cycle [23,42,43]. The release of VIP from the neurons of the SCN regulates the biosynthesis of melatonin in the pineal gland [44]. The synergistic coactivation of VIP and GABAergic pathways in the brain has been identified as a key step in priming the molecular oscillation responsible for the circadian rhythm [22]. Thus, inactivation or defects in the VIP neuronal pathway appears to be a key determinant of the circadian rhythm dysfunction seen in many pathogenic conditions resulting from GABAergic dysfunction, including HD.

## 3. Neuropathogenic Input of Abnormal Regulation of Clock Genes in HD

HD patients have been found to display decreased activity during the day time as they show increased activity during the night-time [6]. Polysomnographic and actigraphic findings in HD patients indicate frequent eye and leg movements during sleep [45]. Several neuroimaging studies of the hypothalamus have revealed prominent neuropathological alterations in the SCN in corroboration with abnormal sleep–wake cycles in HD [6]. Aziz NA et al. reported that there is a delay in the release of melatonin from the pineal gland in HD patients due to abnormal neurotransmission in the SCN [46]. The drosophila model of HD has been seen to exhibit sedentary behaviors as a reflection of impaired circadian rhythm [47]. Experimental data gathered from the sheep model of HD reveals that the sleep disorder resulting from abnormal circadian rhythm is an early sign of the onset of the disease [7]. Kuljis DA et al. indicated that the expression of the mutant *HTT* gene in the brain is responsible for sleep disorder in a bacterial artificial chromosome-based transgenic mouse model of HD [48,49]. In addition, the R6/2 mouse model of HD exhibits progressive disruption in the circadian rhythm leading to reduced physical activity and sluggish behavior [50]. Loh DH et al. observed the progressive deterioration of motor function in association with altered sleep patterns due to defects in the circadian rhythm in the Q175 mouse model of HD [19]. Furthermore, experimental subjects with HD have been reported to display depression and progressive forms of memory deficit resulting from an abnormal circadian rhythm [51]. Considering the aforementioned facts, insights into the mutant HTT protein-mediated dysregulation of circadian clock genes pathway in HD has become an important scientific quest.

Circadian rhythms have been known to be regulated by key clock genes such as Period1 and 2 (Per1/2), Cryptochrome1/2 (Cry1/2), Brain and muscle Arnt-like protein 1 (Bmal1), and Circadian Locomotor Output Cycle Kaput (CLOCK) [52–54]. Bmal1 functions as a transcriptional activator in heterodimeric form in the cytoplasm and it enters the nucleus and binds with the promoter region of Per and Cry, called the enhancer box (E-box), to regulate the expression of various genes [55,56]. Recently, a gene knockout study in embryonic stem cells (ESCs) indicates that the Bmal1/CLOCK gene regulates the transcription of REV-ERB$\alpha$/$\beta$, which plays an important role in neuronal growth, lipid metabolism, and inflammatory processes [57]. Cry1 and Bmal1/CLOCK also modulate the feedback loop of D-box binding protein and interleukin-3-regulated protein which is also important for the regulation of neuroplasticity [58,59]. Clock genes have also been involved in non-circadian phenotypes such as the regulation of immune cells, metabolic pathways, and their loss of function which leads to abnormal aging and the progression of malignant disorders [52]. Notably, the genetic ablation of the Per gene in the drosophila model has been reported to induce mitochondrial dysfunction and oxidative stress, leading to prominent neurodegeneration in the brain [60]. In addition, the Per mutant mouse model has been reported to display abnormal mitotic events due to defects in tumor suppressor genes, thereby indicating the roles of circadian clock genes in cell-cycle control [61]. In addition, an experimental mouse model with the conditional deletion of the Bmal1 gene in the excitatory forebrain neurons has been reported to exhibit cognitive impairments [62]. Furthermore, several experimental studies reported that the aberrant expression or dysfunction of clock genes leads to cognitive impairment, movement, and mood-related disorders in many neurodegenerative conditions, including HD [63]. Circadian rhythm abnormalities following sleep disruption appear to be the prominent clinical manifestation of human subjects, as well as many experimental models

of Alzheimer's disease (AD) and Parkinson's disease (PD) [64]. In addition, the abnormal regulation of clock genes has been identified to be associated with various neuropsychiatric manifestations observed in autism spectrum disorder (ASD), attention-deficit/hyperactivity disorder (ADHD), major depressive disorder (MDD), bipolar disorder (BP) and schizophrenia (SCZ) [53].

Ample reports indicate that expression of mutant *HTT* gene alters the circadian rhythms often before the appearance of involuntary movements in HD [6]. Sleep during night time appears to be progressively reduced and fragmented as neurological symptoms of HD progress [7]. Moreover, abnormal circadian rhythms have been reported to aggravate the progression of the clinical symptoms of HD [65]. HD has been characterized by dysfunctions in the transcriptional regulation of clock genes, which in turn are considered to be an initial trigger for various neuropathogenic changes and mental illnesses [6,51,66]. During the early stage of pathogenesis, HD displays various neuropsychiatric symptoms such as depression, anxiety, stress aggression, psychosis, apathy, obsessive-compulsive behaviors, and psychosis [67,68]. These neuropsychiatric symptoms are multifactorial in origin and are known to be associated with sleep disruption resulting from abnormal circadian rhythmicity [67]. Thus, it can be speculated that the dysregulation of clock genes might be an early pathogenic molecular event prior to the obvious motor and behavioral manifestation of HD. However, the initiation of the abnormalities in the regulation of clock genes upon the pathogenic onset in HD, and the molecular mechanisms by which mutant HTT proteins impair their functions, remain largely unknown.

The abnormal sleep patterns noticed in the fly model of HD have been reported to be linked with an alteration in the transcription of clock genes [18]. Abnormal sleep–wake disorders noticed in the R6/2 mouse model of HD have been reported to be associated with aberrant expressions of Per2 and Bmal1 in the striatum and SCN [50,51]. Moreover, R6/2 mouse models have also been characterized by low levels of VIP expression and its receptor VIPR2 in the brain [69]. Alteration in the metabolic events in the liver of R6/2 mice has been reported to be associated with the abnormal expression of Cry1, D site of albumin promoter binding protein (DBP), and Per2 [50]. Further, the Bmal1 knockout mouse has been characterized by gliosis, neuronal loss, the degeneration of presynaptic terminals, and decreased neural connectivity upon the 3-nitropropionic acid-induced acute HD condition [70,71]. Notably, the supplement of sleeping pills in the R6/2 mouse has been reported to revert the function of Per2 resulting in a significant improvement in cognitive performance [72,73]. While the involvement of clock genes in neuroplasticity has been increasingly noticed, prolonged sleep disruption and the expression of the mutant *HTT* gene have been known to interfere with the regulation of neuroregenerative plasticity [74] (Figure 2). Therefore, regulation of neurogenesis in stem cell niches of the brain can be expected to be linked with the expression of the clock genes.

In a healthy brain, the normal secretion of melatonin and serotonin takes place to ensure the proper sleep–wake cycle and neuroplasticity. In the HD brain, an imbalance in secretions of melatonin and serotonin is responsible for abnormal circadian rhythms and leads to depression, cognitive deficits, and impaired neurogenesis.

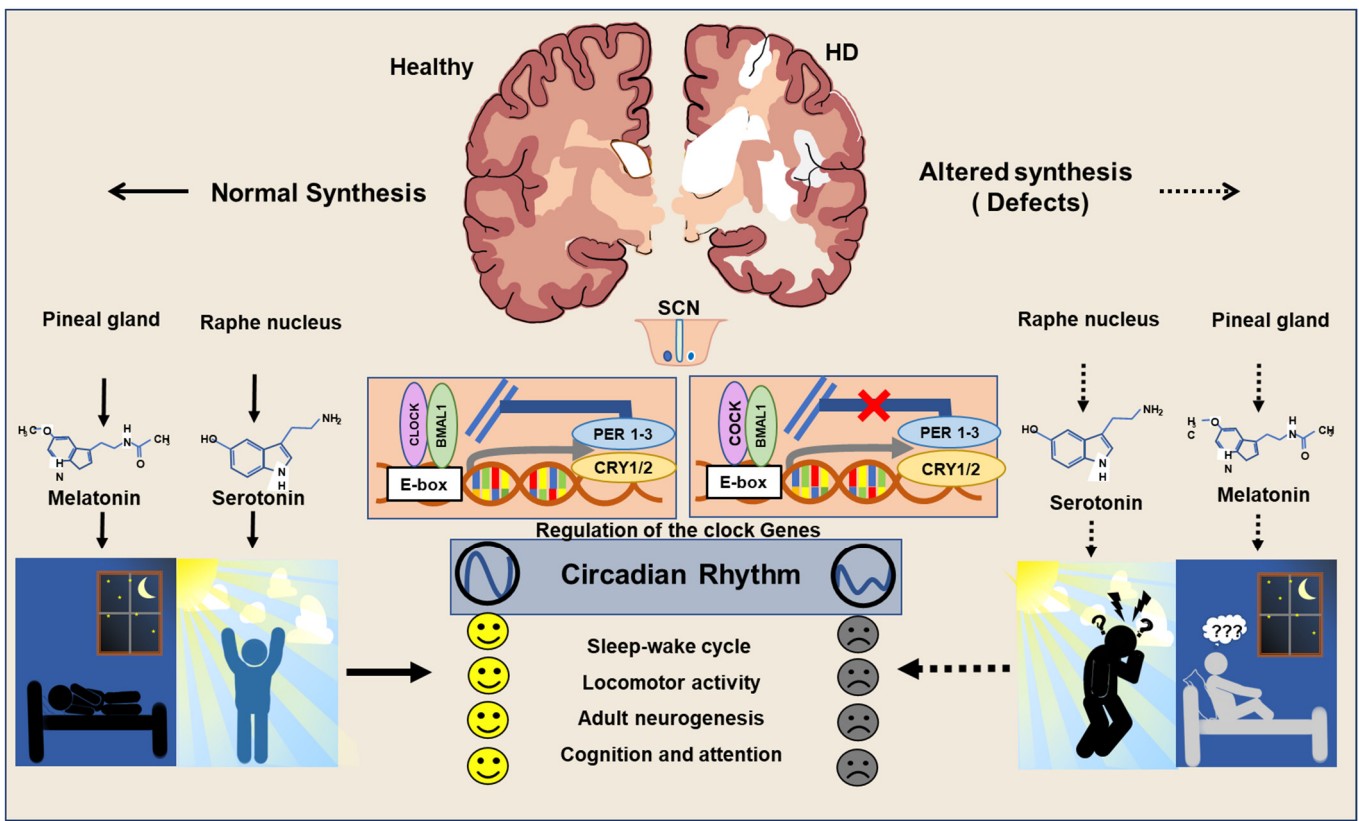

**Figure 2.** Regulation of circadian clock in Healthy and HD brains.

## 4. Potential Overlap between Altered Clock Gene Pathway and Dysregulation of Neuroregenerative Plasticity in HD

Experimental evidence indicates that mitosis of NSCs in the adult brain mostly occurs at night in experimental animals due to their nocturnal behavior [21,75]. HD has extensively been characterized by aberrant neurogenesis in the hippocampus, striatum, and olfactory bulb (OB) [11–13,15,76]. Abnormal neurogenic events such as stem cell quiescence, reactive neuroblastosis, and defect in maturation and integration of new-born neurons have been well documented in the experimental models of HD [11–15]. The occurrence of adult neurogenesis has also been recognized in other brain regions including the cortex, striatum, amygdala, spinal cord, hypothalamus, and the brain stem [77,78]. Among them, understanding the regulation and functional roles of neurogenesis in the hypothalamus has become increasingly important as it appears to take part in many neurophysiological pathways that overlap with many physiological functions [79]. However, direct evidence for the regulation of neurogenesis in the hypothalamus in HD remains limited. While regulation of neurogenesis in the hippocampus has been known to be drastically affected by many neurodegenerative diseases including HD, reports on the modulation of neurogenesis in the hypothalamus under the influence of pathogenic conditions remain limited. An immunohistochemical study by Gabery S et al. revealed that the decline in the number of vasopressin- and oxytocin-expressing neurons noticed in the hypothalamus of the post-mortem human HD brain might be due to mutant *HTT*-mediated impaired neurogenesis [80]. Depression and anxiety-like behaviors have been reported to occur in subjects with HD due to the dysregulation of neurogenesis and abnormal neuroplasticity in the hypothalamus [81,82]. Hypothalamic neurogenesis has been noticed to be controlled indirectly by clock genes such as Bmal1, Per1, and Per2, while the glial differentiation was found to be high in neurospheres derived from mice lacking Cry1 and Cry2 genes [21]. The Bmal1 deficient mice show premature aging, neurodegeneration, and cognitive deficits along with a reduced level of NSC proliferation and the impaired migration of neuroblasts in the brain [83]. Reduced cell proliferation and a lower number of secondary neurospheres have

also been observed from the NSC isolate with the absence of both Cry1 and Cry2 circadian clock proteins [21]. Moreover, the deletion of Per2 has also been reported to induce the cell cycle exit of NSCs [84]. Thus, defects in the regulation of the clock gene pathway seen in HD can be attributed to aberrant neurogenesis in different brain regions, including the hypothalamus. Therefore, correcting the defective clock gene pathway could be one of the valid strategies to boost neuroregenerative plasticity, thereby compensating for neurodegeneration in HD. Recently, optogenetics and chemogenetics-based technologies have been considered to have an advantage to modulate the aberrant clock gene pathways in the brain [85]. Therefore, the implementation of tailored scientific strategies that rectify the clock gene pathway and promote neuroregenerative plasticity might provide a valid treatment option for HD (Figure 3).

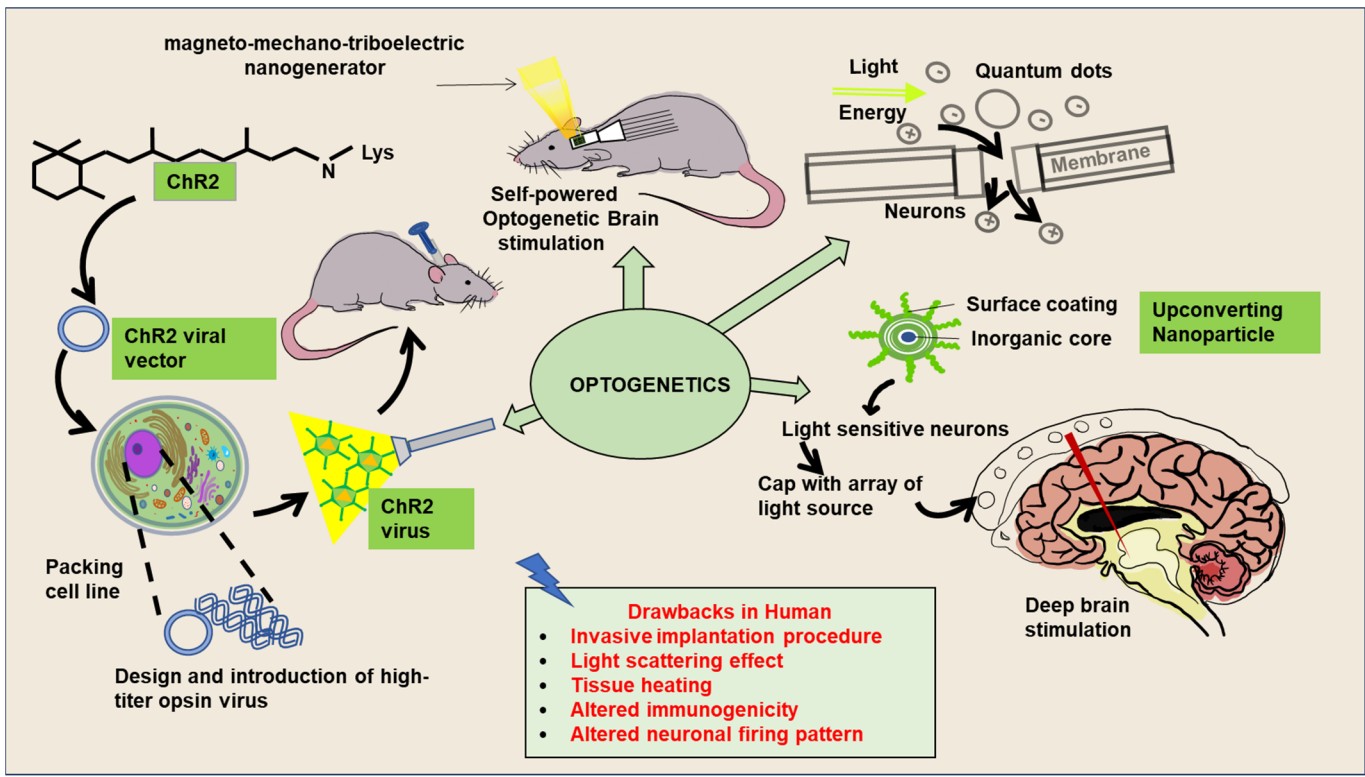

**Figure 3.** Optogenetic modulation of neurons in experimental brains.

The graphical illustration describes various optogenetic approaches to modulate neurons in experimental brains. A magneto-mechano-triboelectric nanogenerator (MMTENG) is inserted into the mouse to stimulate neurons. Channelrhodopsin (ChR)-2 viral vector injected into the brain which is modulated by blue light led to neuronal excitation. The introduction of quantum dots (QDs) and upconversion nanoparticles (UCNPs), which are illuminated by near-infrared spectroscopy (NIR) lights into the deep brain region, can stimulate neuronal activity.

## 5. An Overview and Significance of Optogenetics and Chemogenetics-Based Experimental Interventions for Neuronal Activities

Optogenetics refers to the cutting-edge technology of implementing light to examine and regulate gene expression, as well as desired cellular functions in intact living systems [86]. A specific wavelength of light has been used to activate or deactivate a subset of neuronal populations that are genetically modified to produce light-responsive proteins called opsins [87]. In particular, the primary light-sensitive opsins such as channel rhodopsin and halorhodopsin have extensively been implemented for experimental exploitation in animals, for which the implantation of a regulatable light delivery device in a specific brain area is mandatory to attain high spatiotemporal resolution [88]. While degeneration of GABAergic medium spiny neurons has been ascertained as the unique underlying cause of HD, selective optogenetic stimulation provoked GABAergic transmission in the somatostatin-positive striatal interneurons in R6/2 mice [89]. Using the optogenetic method, Cepeda et al. revealed that abnormal GABAergic transmission in the striatum of the transgenic animal models of HD occurs due to defects in multiple neuronal populations [90]. Another study found that wireless optogenetic stimulation of GABAergic neurons upon electrographic detection of spontaneous hippocampal seizures resulted in shorter seizure durations in patients with temporal lobe epilepsy [91]. Though optogenetics can be proposed to correct the abnormal GABAergic transmission in HD, optogenetic methods have been associated with certain practical difficulties when it comes to an implication in humans as it involves neuro-invasive experimental procedures, such as the implantation of optical fiber into the brain. The effective operation of the optical fiber generates heat that could be detrimental to the tissue [92]. Eventually, penetration of light to the region located deep in the brain tissue is limited by the scattering effect and this issue may be overcome by using longer wavelengths of light [93] (Figure 3). Yet another limitation of optogenetics is the non-specific transfection of optogenes in neurons and the fact that implantation of optrodes requires prolonged anaesthesia that could also lead to some adverse effects [94]. Therefore, the recent advancement in scientific strategies such as chemogenetics appears to have offer alternative to optogenetics as it overcomes the key limitations associated with conventional invasive neurosurgeries.

Chemogenetics involves the administration of inert exogenous ligands that selectively target the non-immunogenic synthetic receptors or enter through ion channels to influence the desired signaling pathways in a given cell type [95]. Unlike optogenetics, chemogenetics does not require the invasive implantation of a light device. Different types of synthetic genetic adducts, cytoplasmic enzymes, and membrane-spanning receptors have been utilized for the operation of chemogenetics [96]. Genetically engineered proteins, such as receptors activated solely by a synthetic ligand (RASSL) and designer receptors exclusively activated by designer drugs (DREADDs), are modified forms of G protein-coupled "designer" receptors that have low affinity for their endogenous ligand, but a high affinity for selective exogenous ligands, such as clozapine-N-oxide (CNO) or salvinorin B (SALB) [95,97,98]. Deschloroclozapine (DCZ) is highly selective for hM3Dq and hM4Di DREADDS with good brain concentration profiles, thus providing good chances for multiplexed/bimodal control of physiological systems along with side-effect-free brain theranostics. These exogenous ligands can be administered locally or systemically to induce or deactivate cellular activities, metabolism, and downstream signalling cascades through the modulation of the activities of DREADDs and RASSL [95]. In particular, the efficacy of different types of DREADDs is currently being pursued to identify the neural pathways related to cognition, motor functions, emotions, drug addiction, and drug abuse in various experimental models, including non-human primates [95,98,99]. For the modulation of neurotransmission in the brain, mutant forms of human muscarinic acetylcholine receptors, such as Gq-coupled human M3 muscarinic DREADD (hM3Dq), human M4 muscarinic DREADD (hM4Di), Gs-DREADD (GsD), Rq (R165L), and kappa-opioid-receptors (KORD), have been used as DREADDs which can be regulated by respective exogenous ligands [100].

Chemogenetics-mediated neuromodulation by stereotactic delivery of adeno-associated virus (AAV) vectors containing hM3D (Gq) or Human synapsin (hSyn)-hM4D(Gi) into different sites of the basal ganglia significantly improved motor performance in a 6-hydroxydopamine (OHDA) injected animal model of PD [101]. A prominent experimental suppression of the hyper neural activities by infusion of Adeno-associated virus (AAV) vectors encoding hM4Di-DREADD into the subarachnoid space and hippocampus, followed by CNO treatment, has been reported to decrease Amyloid-β aggregation in a transgenic animal model of AD [102]. Functional neuroimaging and behavioral studies revealed that injection of an adenoviral vector carrying a hM3Dq DREADD into the vitreous-induced abnormal neural activates and anxiety-like symptoms in association with altered signatures of circadian rhythm in Opn4Cre/+ mice expressing Cre-dependent melanopsin in the RGCs [103]. While synthetic ligands have been known to cross the blood–brain barrier (BBB), intracranial injections of recombinant AAV vectors encoding Gq-DREADDs have initially been considered for selective neuronal transfection in the human brain [104]. To overcome stereotactic injection-related adverse issues, non-invasive AAV delivery by microbubble-enhanced focused ultrasound (FUS) waves used at specific locations in the brain has been proposed [105]. In addition, a non-invasive in vivo retrograde gene delivery strategy modulates neuronal subpopulation in the brain from the periphery, using AAV vectors encoding chemogenetic receptors [106]. Furthermore, in chemogenetic platforms, positron emission tomography (PET) has been used for the non-invasive measurement of the expression and anatomic site of chemogenetic receptors, as well as the detection of radiolabeled clozapine and other ligands [28]. The chemogenetic approach has an advantage over deep-brain stimulation, which is presently used to treat symptoms of parkinsonism, as it eliminates the need for a permanent stimulating electrode implant while maintaining scalable control over neuromodulation via the dosage of the chemogenetic effector drug [28]. Considering the aforementioned facts, FUS and retrograde gene delivery strategy-based chemogenetics can be non-invasively implemented to restore the defects in the circadian rhythm (Figure 4).

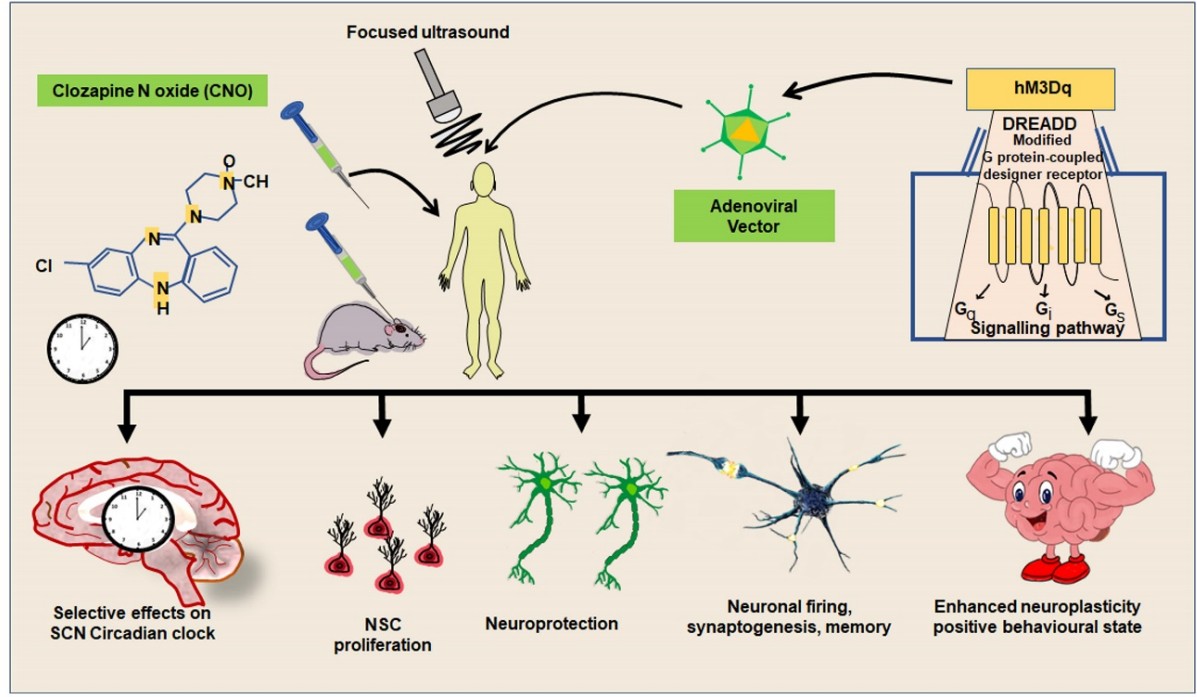

**Figure 4.** The regulation of neuronal activity by chemogenetics.

The figure indicates the different strategies of chemogenetics. Designer receptors exclusively activated by designer drug (DREADD) based human M3 muscarinic (hM3) receptor (hM3Dq) are inserted into the viral vector and surgically injected into the brain. The entry of DREADD into specific brain regions can be improved by focused ultrasound. The DREADD-infected neurons can be activated by clozapine N-oxide (CNO), which is given orally or via injection. This can result in increased neuronal firing, neural stem cell (NSC) proliferation, and neuroplasticity, thereby improving mental health and behavioral outcomes along with selective effects on the circadian clock.

## 6. Discussion

Chemogenetic-innervation of VIP neurons as a subtle therapeutic strategy to rectify aberrant clock gene pathways and GABAergic system to boost neurogenesis in HD.

GABAergic synaptic input plays a significant role in refining and regulating circadian rhythms in the SCN [24,107]. Interestingly, VIP neurons that are present in the SCN receive direct glutamatergic input from the RGCs of the retina and transmit the photic inputs to other neurons through the co-secretion of VIP and GABA [108]. VIP has been known to synchronize the circadian rhythm by regulating the expression of the core clock genes, Per1 and Per2, in the SCN of the brain [19]. Similarly, GABA appears to play a crucial role in the regulation of clock gene expression in the SCN [24,107]. While VIP has been reported to enhance the synaptic transmission of GABA in different areas in the normal brain, the expression of mutant *HTT* gene has been reported to decrease the synthesis of VIP, thereby leading to a defect in the GABAergic input in the brain of subjects with HD [69]. To note, an in situ hybridization study of the post mortem human HD brain indicated reduced transcriptional levels of VIP in the SCN [109]. Considering the aforementioned facts, it can be presumed that the decreased levels of VIP responsible for abnormal GABAergic neuro-transmission in the SCN of the brain might be an underlying basis for the dysregulation of the circadian rhythm. Recent experimental evidence strongly advocates that VIP expression prevents neurodegeneration by mitigating microgliosis and producing neurotrophic factors in the brains of neurodegenerative experimental models [110,111]. Therefore, the restoration of the VIP-mediated signaling pathway in the brain might mitigate the sleep disorders seen in many diseases, including HD. The VIP-expressing neurons have also been known to co-express the muscarinic acetylcholine receptors [112,113]. Therefore, the implementation of chemogenetic receptors such as hM3Dq-DREADD can be used to activate the VIP neurons in the SCN, through which the aberrant GABAergic inputs and dysregulation of the clock gene pathway in the SCN can be restored in HD. Many studies have demonstrated the implementation of chemogenetics in pre-clinical models of neurological disorders, including PD and epilepsy [114,115]. The ligands of chemogenetic approaches such as clozapine and CNO are clinically approved for human trials [116]. AAV vectors are approved for phase I and phase II clinical trials by the Recombinant DNA Advisory Committee and the Food and Drug Administration (FDA) [117]. While the chemogenetic regulation of neural transmission has been an intense scientific focus, the implementation of recombinant AAV vectors encoding Gq-DREADDs for the regulation of circadian rhythm in HD has not been proposed yet. Further, the proposed chemogenetic approach to re-establish the circadian rhythm via the activation of VIP neurons can be expected to facilitate enhanced neurogenesis in the brains of the subjects with HD. With careful use, chemogenetics-based treatment would be expected to provide a potent spatial and temporal resolution for the exact manipulation of the brain functions related to specific behavior. Thus, the chemogenetic approach can effectively be translated to treat sleep disorders, movement disorders, and dementia in HD (Figure 5).

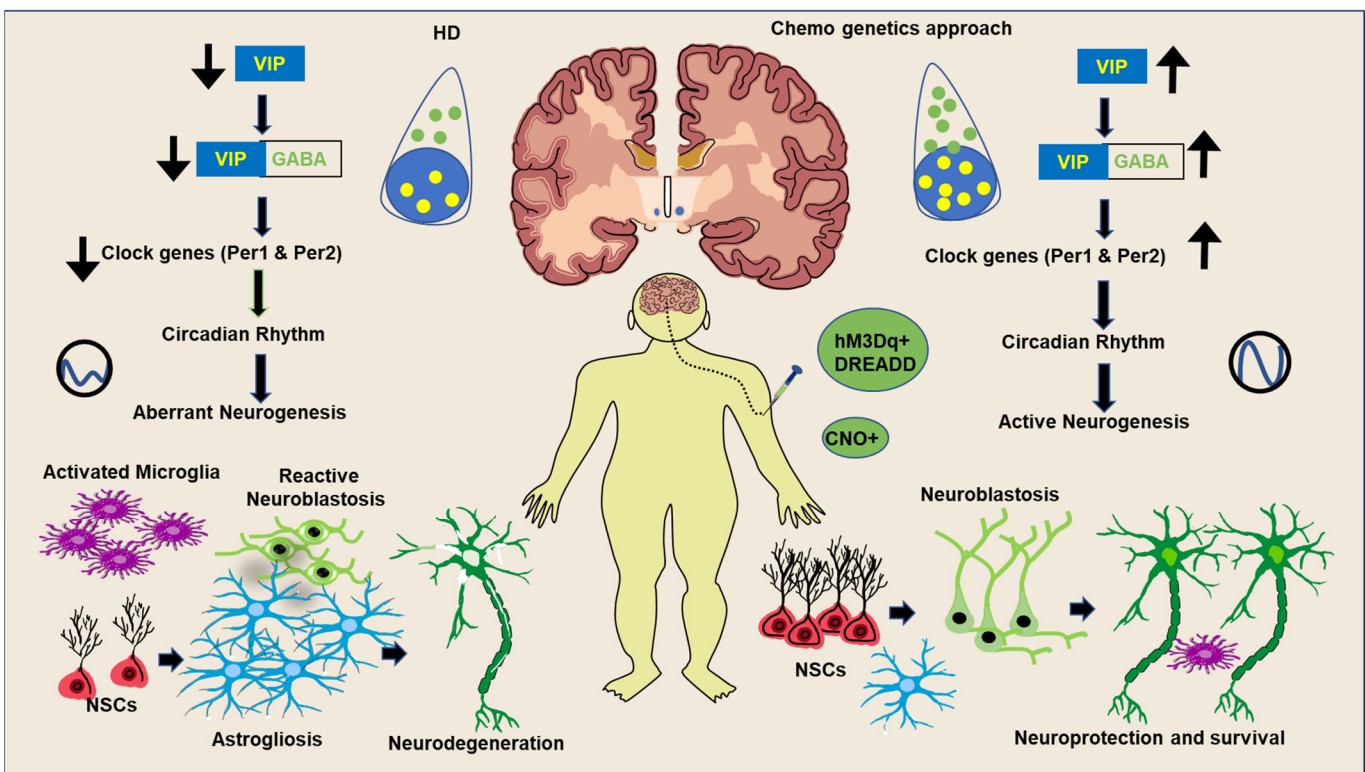

**Figure 5.** Possible therapeutic implication of chemogenetics in HD.

The graphical representation illustrates the neuropathological abnormalities and defects in the vasoactive intestinal peptide (VIP) neurons and gamma aminobutyric acid (GABA)-ergic system, abnormal expression of clock genes, and aberrant hypothalamic neurogenesis in HD (Right side). The figure indicates the hypothesis that the activation of VIP neurons by chemogenetics approach through the retrograde route might restore neuroregeneration via GABAergic inputs in HD (Left side).

## 7. Conclusions and Future Directions

At present, the progressive pathogenic progression of HD appears to be genetically unstoppable by any sort of treatment or medication. The currently available medications and treatments are useful for the management of the clinical symptoms, while there is no complete cure established for HD. While the treatment strategy that suppresses the expression of mutant *HTT* gene is underway, the available gene-modifying techniques appear to alter the expression of wild-type *HTT* gene leading to further adverse effects. Thus, targeting the pathogenic consequences resulting from mutant *HTT* gene and boosting the neuroregenerative potentials of NSCs can be highly beneficial at the moment. The abnormal expression of mutant *HTT* gene induces hypothalamic pathology through the aberrant expression of clock genes. The altered circadian rhythm appears to be associated with the disturbed sleep–wake cycle, hormonal imbalance, and depression which lead to a considerable decline in cognitive performance, eventually leading to impaired neuroregenerative plasticity. Recent advancements in chemogenetic and optogenetic-based approaches represent a conceivable strategy to rectify the aberrant expression of clock genes in the brain. While defects in VIP neurons appear to be a key pathogenic event associated with abnormal GABAergic transmission, implementing chemogenetics could be a promising therapeutic attempt to enhance the activity of the VIP neurons in the hypothalamus, as well as the hippocampus, by which it could be possible to realign the clock genes and the circadian rhythm that are impaired in HD. Eventually, this approach could effectively aid in reinstating neuroregenerative plasticity in HD, as the expression of clock genes facilitate the cell-cycle regulation of NSCs, provoking neurogenic process in the brain. Like other

treatments, limitations that arise from unknown adverse effects related to chemogenetics could not be ignored completely. Thus, the implementation of chemogenetics under proper safety guidelines would be an achievable target in humans. The proposed approach could be translated to treat many other human diseases that are connected to abnormal sleep–wake cycles and aberrant neuroregenerative plasticity.

**Author Contributions:** M.K. and V.T. conceived the idea, and hypothesis and contributed to the framework of the manuscript and illustrations. S.R., R.S., J.F.V.A., S.M.D., D.B.S., V.T. and M.K. wrote the initial draft. All authors contributed to the entire revision of the article and made critical comments and suggestions. All authors have read and agreed to the published version of the manuscript.

**Funding:** M.K. gratefully acknowledges a research grant (SERB-EEQ/2016/000639), an Early Career Research Award (SERB-ECR/2016/000741) from the Science and Engineering Research Board (SERB) under the Department of Science and Technology (DST), Government of India. V.T. gratefully acknowledges research funding from the Department of Science and Technology, Nanomission, Government of India (Grant No. DST/NM/NB/2018/10(G)), Science and Engineering Research Board, Department of Science and Technology, India [Grant No. YSS/2014/00026] University Grants Commission, India [Grant No. F. 4-5(24-FRP)/2013(BSR)]. M.K. acknowledges MOE-RUSA 2.0 Biological Sciences, and V.T. acknowledges MOE-RUSA 2.0 Physical Sciences, Bharathidasan University for their financial support. J.F.V.A was supported as a project assistant from the project grant SERB-EEQ/2016/000639. R.S. was supported as JRF and SRF from the SERB (YSS/2014/00026). Divya Bharathi Selvaraj is the recipient of the RUSA 2.0 project fellowship (Ref. No. BDU/RUSA 2.0/TRP/BS/Date 22/04/2021).

**Institutional Review Board Statement:** Not applicable.

**Informed Consent Statement:** Not applicable.

**Data Availability Statement:** Not applicable.

**Acknowledgments:** M.K. and V.T. have been supported by the University Grants Commission-Faculty Recharge Programme (UGC-FRP), New Delhi, India. The authors would like to thank Anaswara TS for skilled assistance in the digital artwork. The authors acknowledge UGC-SAP and DST-FIST for the infrastructure of the Department of Animal Science and Department of Chemistry, Bharathidasan University.

**Conflicts of Interest:** The authors declare that there are no conflict of interest.

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
