# Peer review of "Intertwining Neuropathogenic Impacts of Aberrant Circadian Rhythm and Impaired Neuroregenerative Plasticity in Huntington’s Disease: Neurotherapeutic Significance of Chemogenetics"

_jmp, doi:10.3390/jmp3040030_

Round 1

Reviewer 1 Report

Reviewer comments to the Authors

·         The reviewed manuscript explains the potential link between mHTT-mediated dysregulation of circadian clock genes and impaired neurogenic events in Huntington’s chorea.

·         the manuscript emphasizes the alternative use of the proposed chemogenetic activation of vasoactive intestinal peptide (VIP)-expressing GABAergic neurons in the brain as a therapeutic strategy to reprogram the clock gene pathways rather than invasive optogenetics methods in mitigating the neurodegenerative failure in HD.

·         The discussion section was relevant to the topic and explained clearly.

·         The manuscript has concluded clearly and precisely.

Overall the manuscript is clear, concise, and well-structured.

·         The authors are asked to address the minor corrections mentioned below:

1.      There is some similarity in the introduction (Mainly the First 5 consecutive sentences) as in the review article “Cell cycle re-entry of neurons and reactive neuroblastosis in Huntington's disease: Possibilities for neural-glial transition in the brain by Nivethita Manickam et al., 2020.

2.      Under the subheading “Regulation of circadian rhythm in the physiological state”, the sentence “VIP acts through VPAC2 receptors to stimulate the secondary messengers cAMP, and PKA signaling cascade and pre-synaptically enhance the release of GABA release in the neuronal population 6 of SCN [36]” has repeated twice.

3.      In figure 1, the abbreviations for RHT, GHT, and IGL has not given in the context and are not discussed also.

4.      The concept behind figure 1 should be discussed for ease of understanding without it, it is quite irrelevant to the manuscript.

5.      Under the subheading “Neuropathogenic input of abnormal regulation of clock genes in HD”, the First and second sentences are not finished completely. In the first sentence, “While HD patients have been found to display decreased activity during day time, they show increased activity during the night time thereby…..”

6.      The scope of future studies that could be initiated based on the conceptual review can be elaborated further.

7.      There are misalignments in the reference section.

Author Response

Reviewer 1

Reviewer comments to the Authors

The reviewed manuscript explains the potential link between mHTT-mediated dysregulation of circadian clock genes and impaired neurogenic events in Huntington’s chorea.

The manuscript emphasizes the alternative use of the proposed chemogenetic activation of vasoactive intestinal peptide (VIP)-expressing GABAergic neurons in the brain as a therapeutic strategy to reprogram the clock gene pathways rather than invasive optogenetics methods in mitigating the neurodegenerative failure in HD.

The discussion section was relevant to the topic and explained clearly.

The manuscript has concluded clearly and precisely.

Overall, the manuscript is clear, concise, and well-structured.

Response: We express our sincere thanks to Reviewer 1 for the careful reading, positive feedback on the structure and all sections of the manuscript, and for suggesting minor corrections. We acknowledge that the clarifications and remarks raised by the reviewer are highly insightful and helpful for the improvement of the content. We have incorporated all the suggestions and clarification of the reviewer. 

The authors are asked to address the minor corrections mentioned below:

  1. There is some similarity in the introduction (Mainly the First 5 consecutive sentences) as in the review article “Cell cycle re-entry of neurons and reactive neuroblastosis in Huntington's disease: Possibilities for neural-glial transition in the brain by Nivethita Manickam et al., 2020.

Response: We thank reviewer 1 for this critical remark. We have rectified the mistakes and revised the overlapping sentences.

  1. Under the subheading “Regulation of circadian rhythm in the physiological state”, the sentence “VIP acts through VPAC2 receptors to stimulate the secondary messengers cAMP, and PKA signaling cascade and pre-synaptically enhance the release of GABA release in the neuronal population 6 of SCN [36]” has repeated twice.

Response: We apologize for the mistake. We have removed the repeated sentence.    

  1. In figure 1, the abbreviations for RHT, GHT, and IGL has not given in the context and are not discussed also.

Response: The figure legends have been improvised. The abbreviations have been properly denoted in the revised version of the manuscript.  

  1. The concept behind figure 1 should be discussed for ease of understanding without it, it is quite irrelevant to the manuscript.

Response: We agree with Reviewer 1. Fig 1 represents the overview of the interlink between brain regions responsible for the regulation of circadian rhythm through light and non-light-based inputs which we consider useful for the introductory section before going into its dysregulation in HD. We have incorporated a segment of text to describe the concept behind figure 1.

  1. Under the subheading “Neuropathogenic input of abnormal regulation of clock genes in HD”, the First and second sentences are not finished completely. In the first sentence, “While HD patients have been found to display decreased activity during day time, they show increased activity during the night time thereby…..”

Response: We apologize for the editing mistake. We have rectified the incomplete sentence.

  1. The scope of future studies that could be initiated based on the conceptual review can be elaborated further.

Response: We thank reviewer 1 for the insightful suggestion, we have included scope and future direction in the revised manuscript.

  1. There are misalignments in the reference section.

Response: The reference section has been updated and realigned.

Reviewer 2 Report

I find this review very useful for a comprehensive interpretation of physiological and neuropathological aspect of SPN VIP mediated axis and clock genes pathway in Huntington's Disease, describing the interplay between aberrant circadian rhythm and impaired neuroregenerative plasticity, especially demonstrated in preclinical models, and offering an insight to the possible application of chemogenetics in rectifying the abnormal circadian clock.

Authors address each aspect first from a physiological perspective and second from pathological evidence in neurodegenerative disorders and in Huntington's Disease (if available as evidence in this field are scarce). 

A major issue I think should be addressed from the authors is the role of chemogenetics application in the present and future scenario of therapetic interventions for Huntington's Disease. How could this treatment change the approach from the ones available nowadays? Should it be considered as a part of a multimodal approach of neurodegenerative disease treatment? Could it interfere with htt expression or mhtt aberrant action?

I think also that authors should better address the role of clock genes in neuropsychiatric manifestations of HD. It is well known that psychiatric disturbances and cognitive impairment have complex neuropathological bases and clock genes regulation of circadian rhythm is not the major determinant of this conditions. 

Another issue that could be pointed is the timing of SPN/clock genes disruption in HD, if this evidence are available. In which stages of the HD course do they present?

Some other minor points I would like to underline:

- in the abstract and in the further paragraphs, please use 'psychiatric symptoms' or similar terms in stead of 'emotional disruption', as more appropriate for HD

- in the introduction section, please substitute irreversible dyskinetic movement with 'abnormal involuntary movements' and 'mood disorders' with 'psychiatric disturbances'

- in the introduction section, please reword this sentence: 'In order to maintain good mental health, sufficient sleep is an essential physiological function'

- in the paragraph 'Regulation of circadian rhythm in physiological state' page 5, the last sentence starting as 'VIP acts through VPAC2 receptors...' seems repetitive with a previous sentence.

- please substitute ref 1 with a more appropriate reference related to the discovery of the gene (i.e. Gusella, et al, A polymorphic DNA marker genetically linked to HD, Nature 1983)

Author Response

Reviewer 2

I find this review very useful for a comprehensive interpretation of physiological and neuropathological aspect of SPN VIP mediated axis and clock genes pathway in Huntington's Disease, describing the interplay between aberrant circadian rhythm and impaired neuroregenerative plasticity, especially demonstrated in preclinical models, and offering an insight to the possible application of chemogenetics in rectifying the abnormal circadian clock.

Response: We extend our sincere thanks to Reviewer 2 for the valuable time and critical reading of the manuscript and clarifications. We consider that the overall feedback and suggestions are highly constructive to improve the clarity of the manuscript.  We have addressed all the clarifications of the reviewer and made a major revision in the manuscript as per the suggestions. 

  1. Authors address each aspect first from a physiological perspective and second from pathological evidence in neurodegenerative disorders and in Huntington's Disease (if available as evidence in this field are scarce).

Response: We thank reviewer 2 for the insightful advice, we have addressed each part from the physiological point of view and discussed the associative pathological changes during neurodegeneration with special reference to Huntington's Disease.

  1. A major issue I think should be addressed from the authors is the role of chemogenetics application in the present and future scenario of therapetic interventions for Huntington's Disease. How could this treatment change the approach from the ones available nowadays? Should it be considered as a part of a multimodal approach of neurodegenerative disease treatment? Could it interfere with htt expression or mhtt aberrant action?

Response:  We thank the reviewer for this important question. At present, the progressive pathogenic progression of HD appears to be genetically unstoppable by any sort of treatment or medication. The currently available medications and treatments are useful for the management of the clinical symptoms, while the is no complete cure established for HD. We believe the proposed chemogenetic approach might be highly relevant to mitigate pathogenic consequences resulting from mutant HTT. It may not directly interfere with the expression of mutant HTT. While the treatment strategy that suppresses the expression of mutant HTT  has been underway, these approaches appear to be associated with interfering with the expression of wild-type HTT itself leading to further adverse effects. We have incorporated a segment of text to address and discuss the aforementioned facts in the revised version of the manuscript.

I think also that authors should better address the role of clock genes in neuropsychiatric manifestations of HD. It is well known that psychiatric disturbances and cognitive impairment have complex neuropathological bases and clock genes regulation of circadian rhythm is not the major determinant of this conditions.

Response:  We thank the reviewer for the suggestions, we admit that regulation of circadian rhythm is a non-motor clinical symptom of HD, while dysregulation of neurogenesis has been linked to the motor, cognitive, and neuropsychiatric manifestations. Ample evidence suggests that clock genes are involved in the regulation of neurogenic proses at the level of neural stem cell proliferation and differentiation which could be overlapped with circadian rhythm. Impaired neurogenesis has been identified as a major pathogenic determinant of psychiatric disturbances and cognitive decline regardless of mHTT-induced neurodegeneration.  We admit the points of the reviewer and we have extended the text to cover the possible roles of clock genes in neuropsychiatric manifestations of HD in the revised manuscript.

  1. Another issue that could be pointed is the timing of SPN/clock genes disruption in HD, if this evidence are available. In which stages of the HD course do they present?

Response: The reports on the timing of SPN/clock genes disruption in HD appears to be limited. However, there exists some evidence that the sleep wakeup cycle has been reported to occur before the onset of motor and cognitive symptoms thereby suggesting disruption of SPN/clock genes could present before the onset of movement disorders in HD. We have included this information in the revised manuscript.

Some other minor points I would like to underline:

- in the abstract and in the further paragraphs, please use 'psychiatric symptoms' or similar terms in stead of 'emotional disruption', as more appropriate for HD

Response: We have replaced the term ‘emotional disruption' with 'psychiatric symptoms' in the abstract and other segments of the manuscript. 

- in the introduction section, please substitute irreversible dyskinetic movement with 'abnormal involuntary movements' and 'mood disorders' with 'psychiatric disturbances'

Response: We have revised the statements as per the suggestion. 

- in the introduction section, please reword this sentence: 'In order to maintain good mental health, sufficient sleep is an essential physiological function'

Response: The sentence has been rephrased.

- in the paragraph 'Regulation of circadian rhythm in physiological state' page 5, the last sentence starting as 'VIP acts through VPAC2 receptors...' seems repetitive with a previous sentence.

Response: We apologize for the error.  We have rectified the text.

- please substitute ref 1 with a more appropriate reference related to the discovery of the gene (i.e. Gusella, et al, A polymorphic DNA marker genetically linked to HD, Nature 1983)

Response: We have placed Gusella, et al, in the reference in the first place.

Round 2

Reviewer 1 Report

Except the corrections in English language, the content of the manuscript is appropriate and can be published in the current form.

Author Response

Reviewer 1 

Except the corrections in English language, the content of the manuscript is appropriate and can be published in the current form.

Response: We sincerely thank reviewer 1 for supporting the manuscript for the  publication.

Senior authors have carefully proofread the entire manuscript. The mistakes and typos have been rectified.

Reviewer 2 Report

Some minor checks for the authors:

- in the abstract, please replace 'altered  fate' with a more readable term (English check)

- in the abstract, please better explain (rewording the present text) the potential link between VIP -expressing GABAergic neurons, circadian rhythm/clock genes, chemogenetics with indirect action on NSC and its role in psychiatric/sleep disturbances associated to HD/other neurodegenerative disorders

- in introduction, please substitute 'For which'

- in the introduction, please rectify ' pathological sleep discord'

Author Response

Reviewer 2

Some minor checks for the authors:

- in the abstract, please replace 'altered fate' with a more readable term (English check)

- in the abstract, please better explain (rewording the present text) the potential link between VIP -expressing GABAergic neurons, circadian rhythm/clock genes, chemogenetics with indirect action on NSC and its role in psychiatric/sleep disturbances associated to HD/other neurodegenerative disorders

- in introduction, please substitute 'For which'

- in the introduction, please rectify ' pathological sleep discord'

Response: We extend our sincere thanks to Reviewer 2 for suggesting some minor changes.

The abstract has been revised and confined. We have incorporated all the corrections in the revised manuscript as advised.